# ZBP1: A Powerful Innate Immune Sensor and Double-Edged Sword in Host Immunity

**DOI:** 10.3390/ijms231810224

**Published:** 2022-09-06

**Authors:** Yu Hao, Bo Yang, Jinke Yang, Xijuan Shi, Xing Yang, Dajun Zhang, Dengshuai Zhao, Wenqian Yan, Lingling Chen, Haixue Zheng, Keshan Zhang, Xiangtao Liu

**Affiliations:** State Key Laboratory of Veterinary Etiological Biology, College of Veterinary Medicine, Lanzhou University, Lanzhou Veterinary Research Institute, Chinese Academy of Agricultural Sciences, Lanzhou 730000, China

**Keywords:** ZBP1, innate immunity, apoptosis, pyroptosis, necroptosis, auto-inflammatory diseases, tumor immunity, signaling transduction, pathogen–host interactions

## Abstract

Z-conformation nucleic acid binding protein 1 (ZBP1), a powerful innate immune sensor, has been identified as the important signaling initiation factor in innate immune response and the multiple inflammatory cell death known as PANoptosis. The initiation of ZBP1 signaling requires recognition of left-handed double-helix Z-nucleic acid (includes Z-DNA and Z-RNA) and subsequent signaling transduction depends on the interaction between ZBP1 and its adapter proteins, such as TANK-binding kinase 1 (TBK1), interferon regulatory factor 3 (IRF3), receptor-interacting serine/threonine-protein kinase 1 (RIPK1), and RIPK3. ZBP1 activated innate immunity, including type-I interferon (IFN-I) response and NF-κB signaling, constitutes an important line of defense against pathogenic infection. In addition, ZBP1-mediated PANoptosis is a double-edged sword in anti-infection, auto-inflammatory diseases, and tumor immunity. ZBP1-mediated PANoptosis is beneficial for eliminating infected cells and tumor cells, but abnormal or excessive PANoptosis can lead to a strong inflammatory response that is harmful to the host. Thus, pathogens and host have each developed multiplex tactics targeting ZBP1 signaling to maintain strong virulence or immune homeostasis. In this paper, we reviewed the mechanisms of ZBP1 signaling, the effects of ZBP1 signaling on host immunity and pathogen infection, and various antagonistic strategies of host and pathogen against ZBP1. We also discuss existent gaps regarding ZBP1 signaling and forecast potential directions for future research.

## 1. Introduction

Recently, Z-conformation nucleic acid binding protein 1 (ZBP1) has become a rising star and attracts a lot of attention [1]. Initially, ZBP1 is identified as an innate immune sensor to activate the IFN-I response and NF-κB signaling, which powerfully resists pathogen invasion [2,3]. Later research pointed to the great importance of ZBP1 to induce PANoptosis, a crossed inflammatory cell death including pyroptosis, apoptosis, and necroptosis [4]. The ZBP1-mediated PANoptosis helps to remove invasive pathogens and uncontrolled tumor cells. However, ZBP1-mediated excessive or aberrant cell death can also cause adverse inflammation in infected or non-infected contexts. Therefore, ZBP1 signaling is a double-edged sword for host immunity. Pathogens need to antagonize ZBP1 signaling to ensure infection, while host needs to regulate ZBP1 signaling to maintain immune homeostasis. In this review, we systematically elucidated the elaborate ZBP1 signaling mechanism and the antagonistic strategies of host or pathogen, which will lay the foundation for developing highly efficient therapeutics for pathogenic infections, auto-inflammatory diseases, and cancer.

## 2. The Expression, Structure and Subcellular Location of ZBP1

ZBP1 was originally found to be significantly upregulated in mouse peritoneal tumor stromal cells or macrophages upon stimulation with IFN-γ or LPS [5]. Later studies revealed that ZBP1 holds different expression levels in different cells [4,6]. As an interferon-stimulated gene (ISG), the 5’ promoter region of ZBP1 contains both the IFN-stimulated gene responsive element (ISRE) and IFN-γ activated site (GAS), so ZBP1 can be induced with IFN-I or IFN-II [2,7,8]. Attributed to the heat shock element (HSE) within the ZBP1 promoter region, the expression of ZBP1 also be increased through heat shock transcription factor 1 (HSF1) when subjected to heat stress [9].

ZBP1 is a member of both the left-handed helical Z-conformation nucleic acid (Z-NA) binding protein family containing the Zα domain and the receptor-interacting serine/threonine protein (RIP) family holding the RIP homotypic interaction motif (RHIM). ZBP1 harbors two N-terminal Z-NA binding domains (ZBD, called Zα1 and Zα2), two intermediate RHIM (RHIM1 and RHIM2), and a C-terminal signal domain (SD) [3,10,11,12] (Figure 1). The N-terminal ZBD enables ZBP1 to sense Z-type or other types of nucleic acid ligands, which is usually critical for the activation of ZBP1 [12,13,14,15]; ZBP1 interaction with other RHIM-containing proteins is the foundation of ZBP1 signaling transduction [3,16,17]. Moreover, the C-terminus SD of ZBP1 is required by the ZBP1-induced IFN-I response [12].

The Zα domains of ZBP1 determine its location in the cell. The initial research on the subcellular localization of ZBP1 indicated that ZBP1 presents the punctate distribution in the cytoplasm and interacts dynamically with stress granules (SGs) and processing bodies (PBs) [18]. SGs and PBs are dynamic non-membrane particles in the cytoplasm used to sequester inactive messenger ribonucleoprotein (mRNP) [19], hinting that ZBP1 is associated with RNA metabolism in cells. Subsequent studies confirmed that not only ZBP1 but also the RNA-editing enzyme ADAR1 and vaccinia virus (VACV) E3L gene-encoded E3 protein (the other two members harboring the Zα domain) all were localized to SGs in stressed cells [20]. In addition, the SGs localization of ZBD proteins depends on the Z-NA sensing of the Zα domain [20]. While the formation mechanism of Z-NA in SGs is unclear, it is sure that the SGs localization of ZBP1 is the result of ZBP1 interaction with the Z-NA [21]. In the process of viral infection, the sensing of ZBP1 to the virus-derived Z-NA also determines its localization. Replication of Influenza A virus (IAV) occurs in the nucleus and produces some incomplete viral gene segments, known as defective viral genomes (DVGs) [22]. The DVGs located in the nucleus likely form virus-derived Z-RNA and are successfully sensed by ZBP1, leading to ZBP1 aggregation and activation in the nucleus [23]. Destruction of ZBP1’s Z-NA binding resulted in the cytoplasmic retention and blocked activation of ZBP1 after IAV infection [23]. Unlike IAV, VACV accumulates the cytoplasmic Z-RNA so that ZBP1 is simply activated in the cytoplasm in response to VACV infection [6]. In summary, ZBP1 shuttles between the cytoplasm and the nucleus and that depends on the location of the nucleic acid ligands recognized by ZBP1. 

## 3. ZBP1 Initiating Innate Immune Signaling

### 3.1. ZBP1 and IFN-I Response

IFN-I expression is mainly induced by various pattern-recognition receptors (PRRs), including Toll-like receptors (TLRs), RIG-I-like receptors (RLRs), cyclic-GMP-AMP synthase-stimulator of interferon gene (cGAS-STING) [24,25]. IFN-I directly induces the ISGs expression, therefore, it is the key to activating innate immunity and inhibiting pathogenic infection [26,27]. Attributing the promoting effect of IFN-I to the maturation of dendritic cells, IFN-I also acts as a bridge between innate and adaptive immunity [28]. 

ZBP1 was initially recognized as a sensor for double-stranded DNA (dsDNA). In mouse embryo fibroblasts (MEFs) or L929 cells in response to immunostimulatory dsDNA (including B-DNA, bacterial and viral DNA) overexpressing ZBP1 significantly increased the IFN-I (IFN-α and IFN-β) expression [2,12]. Knockdown of ZBP1 also notably decreased B-DNA or Pseudorabies virus (PRV)-stimulated IFN-β activation in PK-15 cells [29]. The binding of ZBP1 to dsDNA required the D3 domain (a region overlapping the RHIM1 domain) [2]. ZBP1’s dsDNA ligands have no sequence specificity, but longer dsDNA fragments are more likely to activate ZBP1. That is because longer dsDNA act as the scaffold factor, linking many ZBP1 proteins to form ZBP1-dimer or even ZBP1-oligomer complex [12]. The artificially induced dimerization of ZBP1 also resulted in the upregulation of IFN-I genes in the absence of dsDNA [12]. Furthermore, the Zα, Zβ, D3, and SD domains of ZBP1 all were indispensable to the full activation of ZBP1 [2,12]. Activated ZBP1 complex recruit and phosphorylate TANK-binding kinase 1 (TBK1) and interferon regulatory factor 3 (IRF3), then p-IRF3 dimer migrates to the nucleus and finally initiates the transcription of IFN-I [2,12]. In addition, ZBP1 possesses many potential serine/threonine phosphorylation sites in the 338 to 366 amino acids region. In addition, the IFN-I induction was severely weakened for the mutant of ZBP1 (S352A), indicating that the phosphorylation of ZBP1 is vital to its function [12]. 

ZBP1 acts as both an ISG and the promoting IFN-I-expression gene, which will form the ZBP1-mediated IFN positive feedback loop and provoke a strong innate immune response in dsDNA stimulated cells. However, subsequent studies pointed to the limitation in the ZBP1-dependent induction of IFN-I. Deletion of the ZBP1 did not affect the B-DNA-mediated IRF3 activation and IFN-β production in A549, HepG2 cells, or mice [30,31,32]. On the one hand, similar to the multiple RNA sensor in cells, the cellular DNA sensors are also diverse. For example, cGAS-STING has similar abilities to ZBP1 in strongly recognizing cytoplasmic dsDNA and inducing the IFN transcription [25]. Which represents the cell functional redundancy and ensures effective innate immune responses during pathogenic infection. On the other hand, the regulation of the ZBP1 signaling is very complex. Ectopic ADAR1 or E3 expression powerfully impaired the B-DNA-mediated induction of IFN-β mRNA in MEFs [12]. Additionally, both ADAR1 and E3 had been confirmed to competitively bind Z-RNA with ZBP1 and to block ZBP1-mediated necroptosis via the homotypic Zα domain [6,33,34]. The negative regulation of ZBP1 signaling may be beneficial to cellular immune homeostasis or viral immune escape [35].

### 3.2. ZBP1 and NF-κB Signaling

NF-κB family (RelA, RelB, c-Rel, p50, IκBα/β/ε, IKKα/β/γ) play a crucial role in regulating inflammation and innate immune [36,37]. NF-κB signaling regulates the expression of proinflammatory cytokines and anti-infection factors, including TNF-α, interleukin-1 (IL-1), IL-6, IL-8, adhesion molecules, cc chemokine ligand 5 (CCL5) [38]. NF-κB signaling also controls other cellular processes, such as cell proliferation, differentiation, and apoptosis [39,40]. 

The ZBP1-mediated NF-κB signaling is mechanically very similar to TLR3. TLR3 is an innate immune sensor that recognizes pathogen-associated dsRNA. TLR3 activates NF-κB signaling via toll-like receptor adapter molecule 1 (TRIF) [41]. TRIF interacts with receptor-interacting serine/threonine-protein kinase 1 (RIPK1) via the homotypic RHIM domains with ZBP1 [42]. TRIF-RIPK1 interaction recruits transforming growth factor β-activated kinase 1 (TAK1), TAK binding proteins (TAB1 and TAB2), and the IκB kinase (IKK) complex (IKKα, IKKβ, IKKγ). The protein complex promotes IKK activation, IκB phosphorylation, NF-κB nuclear translocation, and induction of antiviral genes [43,44]. ZBP1 also interacts with RIPK1 and mediates similar NF-κB signaling dependent on their shared RHIM domains [3,16]. However, there is one difference between ZBP1-mediated and TLR3-TRIF-mediated NF-κB signaling. Receptor-interacting serine/threonine-protein kinase 1 (RIPK3), the fourth RHIM-containing host protein, blocks TRIF-RIPK1 signaling to NF-κB by competing with RIP1 for binding to the RHIM of TRIF [41]. For ZBP1-RIPK1 signaling to NF-κB, RIPK3 collaborates with RIPK1 to facilitate ZBP1-mediated NF-κB activation [3,16]. The reasons for the differential regulation of RIPK3 need to be further studied to clarify.

## 4. ZBP1 Initiating Panoptosis Signaling

After suffering from stress or microbial infection, the immune system of cells will initiate programmed cell death (PCD) to maintain homeostasis or obliterate infected cells [45]. Major PCD includes pyroptosis, apoptosis, and necroptosis, and their molecular mechanisms have been elucidated. Apoptosis depends on the activation of the initiator caspases-8 (CASP8), CASP9, CASP10, and the executioner CASP3 and CASP7 [46]. Apoptosis results in the formation of apoptotic bodies and is defined as non-inflammatory cell death [47]. Different from apoptosis, pyroptosis and necroptosis both are characterized by pore-formation and rupture in the plasma membrane [47,48]. Pyroptosis is executed by activated inflammasome to cleave Gasdermin-D (GSDMD) and the pro-IL-1β, pro-IL-18 [49]. The N-terminal pore-forming domain of cleaved GSDMD forms the GSDMD-polymerized membrane pores and pyroptotic cell death [50]. Similar to pyroptosis, canonical necroptosis forms the membrane pores through the RIPK3-mixed lineage kinase domain-like protein (MLKL) necrosome downstream of TNFα-RIPK1 or TLR3/4-TRIF signaling [51,52,53]. The GSDMD pores or MLKL pores both enable intracellular cytokines and damage-associated molecular patterns (DAMPs) to release outside the cell, causing inflammatory cell death [48]. Apoptosis, pyroptosis, and necroptosis are not isolated from each other but are closely linked. The new concept, PANoptosis, is created as an inflammatory PCD pathway and reflects the crosstalk and co-regulation between apoptosis, pyroptosis, and necroptosis [54]. PANoptosis is driven by the PANoptosome complex, the central molecular scaffold to synchronously activate apoptosis, pyroptosis, and necroptosis [55,56]. 

ZBP1 act as a master regulator of PANoptosis and plays a vital role in initiating the assembly of PANoptosome [54] (Figure 2). ZBP1 PANoptosome consists of ZBP1, RIPK3, MLKL, RIPK1, FAS-associated death domain protein (FADD), CASP8, NLR family pyrin domain containing 3 (NLRP3), apoptosis-associated speck-like protein containing a CARD (ASC), and CASP1 [57]. The homotypic/heterotypic interactions between ZBP1 PANoptosome members provide a molecular backbone of ZBP1 PANoptosome formation [56]. Upon virus-derived or endogenous Z-NA ligation, ZBP1 binds to the RHIM domains of RIPK3 and RIPK1 [23,58]. Kinase activity-dependent RIPK3 recruits and phosphorylates MLKL to compose the necrosome and implement necroptosis [6,23,51]. For another, RIPK1 binds with FADD via shared death domain (DD) and the kinase activity of RIPK1 is dispensable for RIPK1-FADD interaction. The conserved Death Effector Domain (DED) in FADD and CASP8 makes FADD interact with CASP8, leading to the auto-processing and maturation of CASP8. The activated CASP8 cleaves executioner CASP3/7 to drive extrinsic apoptosis [59,60]. In addition, ZBP1 PANoptosome also further facilitate the assembly and activation of NLRP3 inflammasome [17,61]. Knockout of ZBP1 fully eliminated NLRP3 inflammasome activation and reduced the release of cytokines upon IAV infection [61]. Further, co-deletion of RIPK3 and CASP8 achieved similar results, indicating the close crosstalk between ZBP1-mediated apoptosis, necroptosis, and pyroptosis [61]. 

Much research elucidated some crosstalk between apoptosis, necroptosis, and pyroptosis, which provides a basis for studying the signaling crosstalk in ZBP1-mediated PANoptosis. CASP8 is the key connection between apoptosis and pyroptosis. FADD/CASP8 are necessary for the activation of CASP1 and cell pyroptosis in response to LPS + ATP, C. rodentium or Yersinia stimulation [62,63,64]. Consistent with that, the heterotypic interaction between CASP8-DED and ASC-PYD results in the recruitment of catalytically inactive CASP8 into the inflammasome complex, thereby activating CASP1 [65,66,67,68,69]. Furthermore, FADD/CASP8 also mediate transcriptional priming of the NLRP3 inflammasome under LPS + ATP treatment [62]. The upregulation effect on NLRP3 transcription may be attributed to the CASP8-FADD-RIPK1 complex activated NF-κB signaling [70,71]. In addition, similar to CASP1, CASP8 act as the executioners of pyroptosis to directly cleave and mature GSDMD and IL-1β [72,73,74,75]. Necroptotic signaling became another node to link necroptosis to pyroptosis. MLKL pores collaborate with GSDMD pores or play the compensatory role in the absence of GSDMD pores to promote the release of inflammatory cytokines and DAMPs [75,76]. In macrophages, MLKL pores-mediated potassium efflux also was pointed to upstream signaling of pyroptosis and initiated the activation of NLRP3 inflammasome in a cell-intrinsic manner [77,78].

## 5. ZBP1 Signaling in Microbial Infection

As a potent innate immune sensor, ZBP1 plays an essential role in immune defense against multiple pathogens infection, including viruses, bacterium, fungi, and parasites (Table 1). For example, ZBP1 detects the incursive virus-derived nucleic acid and destroys the viral replication niche by initiating cell death. In turn, some pathogens have developed diverse tactics to block the ZBP1 signaling to ensure efficient infection. 

### 5.1. Herpesviruses

Members of the herpesvirus family are classified as large dsDNA viruses. Based on genome sequence homologies, host or cell tropism, and replication strategies, herpesvirus is divided into alpha-, beta, and gamma-herpesviruses [99].

Cytomegalovirus (CMV) is β-herpesvirus and leads to extensive clinical symptoms, such as scongenital infection, retinitis, and hepatitis. Mouse cytomegalovirus (MCMV) encodes several cell death suppressors including the M38.5-encoded viral mitochondrial inhibitor of apoptosis (vMIA), the M36-encoded viral inhibitor of caspase-8 activation (vICA), and the M45-encoded viral inhibitor of RIP/RHIM activation (vIRA) [100]. The vIRA of MCMV possesses an N-terminal RHIM domain and interplays with RHIM-containing host protein to block the host’s RHIM signaling [79,80] (Figure 3). Compared with the wild-type MCMV (MCMV^WT^), the M45 mutant MCMV (MCMV^M45mutRHIM^) infection-induced RIPK3-dependent but RIPK1/TRIF-independent necroptosis and show weaker replication in 3T3-SA or SVEC4-10 cells [81]. Subsequent research confirmed that ZBP1 acts as the initiator of RIPK3-mediated necroptosis, and deletion of ZBP1 or RIPK3 both rescue infected cell viability and MCMV^M45mutRHIM^ virulence [4,81]. The Zα2 of ZBP1 is indispensable for sensing the MCMV-derived nucleic acid ligands, and RHIM1 of ZBP1 is required to interact with RIPK3 [82,83]. Intriguingly, ZBP1-mediated necroptotic signaling requires viral immediate-early protein 3 (IE3), an essential protein required for early and late gene transcription, indicating that MCMV-derived RNA but not DNA became the ligand of ZBP1 [82,83]. The vIRA of MCMV^WT^ reshapes the ZBP1-RIPK3 complex and impedes the necroptotic signaling to maintain viral normal replication [101]. In addition, ZBP1 mediates the IRF3-dependent IFN-β transcription and ISG expression in Human cytomegalovirus (HCMV) infected THF cells [102,103]. Knockdown of ZBP1 impairs the IFN-β transcription and overexpression of ZBP1 markedly inhibits replication of HCMV in THF cells [102].

Herpes simplex virus-1 (HSV-1) and varicella zoster virus (VZV) both belong to the α-herpesvirus. HSV-1 is known to induce cytolytic death and cause orofacial infections or encephalitis [100]. Mice BMDMs infected with HSV-1 form a huge PANoptosome consisting of absent in melanoma 2 (AIM2), ZBP1, pyrin, ASC, CASP1, RIPK3, MLKL, RIPK1, FADD, and CASP8 [84]. The PANoptosome efficiently suppresses HSV-1 replication and saves infected animals from death via driving apoptosis, pyroptosis, and necroptosis in macrophages. AIM2, an inflammasome sensing dsDNA, acts as the initiator of the PANoptosome. Knockout of AIM2 in BMDMs completely abolishes CASP1/3/7/8 cleavage, the release of IL-1β and IL-18, and the phosphorylation of RIPK3 and MLKL. In the AIM2-mediated PANoptosome, ZBP1 and pyrin are placed at the crucial point downstream of AIM2. Because separately deletion of ZBP1 or pyrin in BMDMs only partially weakened the CASP1/3/7/8 cleavage, release of IL-1β and IL-18. Only collective loss of ZBP1 and pyrin in BMDMs prevents the HSV-1-induced cell death-like AIM2^−/−^ BMDMs [84]. Consistently, knockdown of ZBP1 in primary murine microglia and astrocytes relieve inflammatory cytokines, neurotoxic mediators, and cell death, which elucidated a new mechanism of HSV-1-induced and ZBP1-dependent neuropathology of the central nervous system (CNS) [104]. Similar to MCMV infection, Zα2 of ZBP1 recognizes HSV-1-derived nascent RNA and RHIM1 of ZBP1 is required to provoke HSV-1-induced necroptosis [85]. HSV-1 encoded ICP6 also contains RHIM domain and RHIM-dependent interacts with ZBP1-RIPK3 complex to inhibit ZBP1-mediated necroptosis in host cells [85,86,87] (Figure 3). Unexpectedly in HSV-1 infected murine cells, the suppression function of ICP6 on necroptosis is invalid and leads to mice mortality [85,87,88]. Besides that, ZBP1 was demonstrated to interact with HSV-1 ICP0 and inhibit the activation of the ICP0 promoter to suppress HSV-1 replication in HepG2 cells [30]. Different from ICP6 of HSV-1, RHIM-containing ORF20 protein encoded by VZV blocks ZBP1-mediated apoptosis rather than necroptosis via forming stable heteromeric amyloid assemblies with ZBP1 [89].

Epstein–Barr virus (EBV) is a chronic human γ-herpesvirus that persists for life. EBV infection causes an increased risk of cancer and autoimmune diseases [105,106]. However, the efficient elimination of EBV is difficult because of the EBV latency after the initial infection. A new hypothesis holds that ZBP1 is not involved in EBV’s lysis replication but is contribute to the EBV latency [107]. There are two possible pathways by that ZBP1 induces the establishment and maintenance of EBV latency and both depends on suppressive dimethylation of the histone H3 lysine 9 residue (H3K9me2) [107]. The first pathway, ZBP1 senses and binds to the strong Z-DNA prone sequences in multiple promoter regions of the EBV genome, then further locates C-terminal binding protein (CTBP) to the Z-sequences within promoters could induce H3K9 methylation by MYC oncogene protein [107,108]. The other pathway, EBV-produced Z-NA activates ZBP1-RIPK1-mediated NF-κB signaling during the initial stages of infection [16]. The unleash p50 homodimers interact with euchromatic histone lysine methyltransferases (EHMT1) to induce H3K9 methylation and the expression silence of EBV genes in viral NF-κB binding sites [109]. The hypothesis may help the EBV eradication by suppressing ZBP1-mediated virus latency but needs to be confirmed by further experiments.

### 5.2. Vaccinia Virus (VACV)

VACV belongs to the poxvirus family and has a large linear dsDNA genome and replicates in the cytoplasm. VACV is used as a vaccine against smallpox and is currently being further developed as an efficient vector for novel vaccines against cancer and other infectious diseases [110,111]. The E3 protein of VACV generates cytoplasmic Z-RNA via its C-terminal dsRNA binding domain (dsRBD), and the virus-derived Z-RNA is sensed and bound by Zα2 of ZBP1 [6]. The interaction between E3-Z-RNA-ZBP1 triggers the ZBP1-RIPK3-MLKL axis-dependent necroptosis to restrain viral replication and alleviate clinical symptoms in VACV-infected mice [6,90]. However, the N-terminal Zα domain of E3 competes with ZBP1 for binding the virus-derived Z-RNA so that ZBP1 is separated from the viral Z-RNA and does not interact with RIPK3 to induce necroptosis [6,90] (Figure 3). The functional substitution of ZBP1 Zα1 (E3^Zα1ZBP1^) or ADAR1 Zα (E3^ZαADAR1^) for E3 Zα still blocks necroptosis and maintains the normal virulence of the VACV, indicating the equivalence of Zα domain of Z-NA binding family in sensing and binding Z-NA and the significance of E3 Zα as a critical suppressor of ZBP1-induced necroptosis [6,91].

### 5.3. Influenza A Virus (IAV)

IAV is a negative sense, single-stranded, lytic RNA virus and belongs to the Orthomyxoviridae family. IAV targets and lyses lung epithelial cells and fibroblasts that cause airway and lung dysfunction as well as cardiovascular diseases [112,113]. IAV infection upregulates the expression of ZBP1 relied on the activated RIG-I-MAVS signaling and IFN regulatory factor (IRF1), and induces the TRIM34-mediated K63-linked polyubiquitination of ZBP1 on the K17 position [114,115,116]. The Zα2 of ZBP1 senses the viral Z-RNA derived from the defective viral genomes (DVGs) (some IAV gene segments produced by the IAV RNA polymerase) in the nucleus [15,23,117]. Subsequently, ZBP1 recruits RIPK1, FADD, CASP8, RIPK3, MLKL, NLRP3, ASC, CASP1, and CASP6 to consist of the ZBP1 PANoptosome and execute PANoptosis [61,118,119] (Figure 2). CASP6 of the PANoptosome serves as the scaffold protein to strengthen the interaction between ZBP1 and RIPK3, and its caspase activity is not required [119]. During IAV infection, ZBP1 PANoptosome-mediated necroptosis differs significantly from TNFα signaling. Attributing to the nuclear localization of ZBP1, RIPK3-MLKL necrosome also migrates to the nucleus so that p-MLKL pores firstly rupture the nuclear membrane and secondly plasma membranes [23]. Although ZBP1-mediated PANoptsis significantly suppresses IAV replication, the inflammatory death of pulmonary cells also causes the lethality of infected mice because of the pulmonary dysfunction and inflammation [61,92,120,121]. Therefore, ZBP1-mediated PANoptosis could be a double-edged sword for IAV-infected hosts. In addition, other members of the Orthomyxoviridae family, such as seasonal strains of IAV and IBV but not the Paramyxoviridae family, also produce the nuclear Z-RNA, and whether ZBP1 also mediates similar PANoptosis against other orthomyxoviruses infection requires further verification [23].

### 5.4. Severe Acute Respiratory Syndrome Coronavirus 2 (SARS-CoV-2)

Since 2019, the Coronavirus disease 2019 (COVID-19) pandemic has led to socioeconomic and psychological distress [122]. COVID-19, caused by SARS-CoV-2, is characterized by acute respiratory distress syndrome (ARDS), multi-organ failure and death [123]. SARS-CoV-2 is a positive sense, single-stranded RNA virus and belongs to β-coronavirus family. High dose infection or late infection with SARS-CoV-2 induce IFN-α/γ responses and the expression of ZBP1 [95,124]. In SARS-CoV-2 infected mice or BMDMs, the additional IFN-β inducing ZBP1 provokes PANoptosis via CASP1/3/7/8, GSDMD, GSDME, MLKL [75,95,125], and deletion of ZBP1 (ZBP1^−/−^) or Zα2 deficiency of ZBP1 (ZBP1^ΔZα2^) substantially alleviated the inflammatory cell death and lethality of mice upon treatment with IFN and SARS-CoV-2 infection [95]. The ZBP1-mediated PANoptosis explains why non-early IFN treatment is not clinically helpful in treating SARS-CoV-2 infection [126,127]. In the early stage of infection, the upregulated ZBP1 via IFN-I intervention induces the appropriate death of infected cells which results in a mild inflammatory response to help host clear the invaded SARS-CoV-2. Nonetheless, later administration of IFN induces the ZBP1-mediated massive inflammatory cell death in the lungs and causes inflammatory cytokine storm, lung dysfunction, and increased mortality [95,124]. The retrospective research which showed the increased IFN levels in serum of patients with serious infection compared with those with moderate COVID-19 also supports the above standpoint [128]. Hence, interfering with ZBP1 function could reverse the negative effect of IFN-I clinical treatment and be an efficient therapeutic strategy against SARS-CoV-2 infection. Moreover, ZBP1 also mediated the same PANoptosis against mouse hepatitis virus (MHV) infection, another member of the β-coronavirus family [95].

Genome analysis reveals the potential Z-RNA prone sequences (called flipons) within the coronavirus genome including SARS-CoV, MERS, and SARS-CoV-2. SARS-CoV-2-derived flipons are mainly present in Nsp2, Nsp12, Nsp13, S, and M genes [129]. That flipons are likely to form Z-RNA during viral replication and transcription and thus activate ZBP1-mediated PANoptosis. Moreover, the Nsp13 helicase of coronavirus contains an identified RHIM and a potential Z-flipon helicase (flipase) activity [129]. Thus, coronavirus Nsp13 could strongly suppress Z-RNA-activated immune response via antagonizing ZBP1-RIPK3 signaling or segregating and unwinding of the Z-RNA [4,129]. In addition, more experiments were needed to confirm the above supposes.

### 5.5. Flaviviruses

Flaviviruses have a positive-sense, single-stranded RNA genome and are generally spread to vertebrate hosts by arthropods. West Nile virus (WNV) and Zika virus (ZIKV) both belong to mosquito-borne flaviviruses and cause encephalitis and severe neurological injuries [130,131]. ZBP1 expression is dramatically increased in the infected murine brain with WNV or ZIKV [132,133]. In WNV-infected mice, lacking ZBP1 leads to elevatory brain viral load, more severe viremia, and significantly higher morbidity and mortality [93]. Transcriptomics analysis shows that apoptosis and pyroptosis pathways are significantly enriched and necroptotic genes (RIP1K, RIPK3, and MLKL) are upregulated in the WNV-infected mice murine brain [134]. Although deletion of ZBP1 inhibited ZBP1-dependent WNV-induced cell death in MEFs, does not cause the viability difference in WNV-infected primary mouse cortical neurons [93]. Similar to WNV infection, ZBP1-RIPK3 also unsuccessfully causes cell death in ZIKV-infected primary mouse cortical neurons [135]. However, ZBP1-RIPK3 effectively provokes necroptosis even though not include apoptosis and pyroptosis in ZIKV-infected human astrocytes, and the necroptosis significantly suppresses viral titers [94]. Hence, flaviviruses may induce ZBP1-dependent cell death only in specific brain cells, such as astrocytes, so that their replication is inhibited. In addition, interestingly, the ZBP1-RIPK1-RIPK3 complex upregulates immune responsive gene 1 (IRG1) to turn cis-aconitate into itaconate in ZIKV-infected primary cortical neurons [135]. Then increased itaconate activates a metabolic state via inhibiting succinate dehydrogenase (SDH) activity and saves the ZIKV-infected mice [135]. That presents a new anti-virus immunometabolic mechanism for ZBP1 in neurons. 

### 5.6. Bacterium

*Yersinia pseudotuberculosis* (a Gram-negative bacterium) infected or lipopoly-saccharide (LPS, a component of Gram-negative bacteria wall)-treated BMDMs arises apoptosis and pyroptosis via the TRIFosome consisting of TRIF, ZBP1, RIPK1, FADD, CASP8, GSDMD [17]. After TLR4 sensing LPS and translocating to the endosome, adapter protein TRIF recruits ZBP1 and RIPK1 via the shared RHIM domains. The TRIF-ZBP1-RIPK1 complex further binds FADD and CASP8, then CASP8 cleaves CASP3/7 and GSDMD to activate apoptosis and pyroptosis, respectively [17,72]. The ZBD of ZBP1 is unimportant for the TRIFosome assembling, indicating that ZBP1 only serves as a scaffold protein to promote the assembly of TRIFosome [17]. Lacking ZBP1 markedly blocks activation of CASP8 and decreases inflammatory death of macrophages, which are not beneficial for host defense against *Yersinia pseudotuberculosis* infection [17,136]. ZBP1-mediated necroptosis in macrophages also play a vital role in LPS-induced lung inflammatory injury [97]. TLR4-TRIF may still be the upstream of ZBP1 signaling because of the TLR4-LPS sensing. In addition to necroptosis signaling, ZBP1 of TRIFosome act as a scaffold to promote TRIF-RIPK1 interaction and M1-ubiquitination of RIPK1, which is important for LPS- or dsRNA-activated inflammatory responses by TLR3/4 [137]. Additionally, knockout of ZBP1 mitigates the release of mtDNA, cytokines, and the activation of the NF-κB pathway in macrophages following LPS stimulation, showing that ZBP1 signaling triggers other innate immune pathways [97]. In addition, *Francisella novicida* (an intracellular Gram-negative bacterium) infected BMDMs form the AIM2 PANoptosome similar to HSV-1 infection [84]. AIM2 functions as the apical sensor to bind the bacterial DNA and initiate the assembly of the PANoptosome [138]. ZBP1 acts as the adapter protein of the AIM2 PANoptosome and is indispensable to provoking PANoptosis [84]. Similar to viruses, some bacterium also develops the capacity to antagonize the ZBP1 signaling. *Enteropathogenic Escherichia coli* (EPEC) encoded type III secretion system (T3SS) effector EspL has a tripartite cysteine protease motif (Cys47, His131, Asp153) that cleaves within the RHIM domains and degrades the RHIM-containing proteins (including ZBP1, RIPK1, RIPK3, TRIF) [139]. The interfering effect of Espl on RHIM signaling could be an important tactic to ensure EPEC infection.

### 5.7. Fungi

*Candida albicans* and *Aspergillus fumigatus* infection also activate ZBP1-dependent PANoptosis and inflammation in BMDMs [98]. Compared to wild-type BMDMs, ZBP1^−/−^ or ZBP1^ΔZα2^ BMDMs show significantly attenuated PANoptosis and the release of IL-18 after fungi infection [98]. While it is not clear how ZBP1 is activated during fungal infection, it is expected that ZBP1-mediated PANoptosis may be important in controlling fungi infection via the inflammatory response [140,141].

### 5.8. Toxoplasma Gondii

*Toxoplasma gondii* belongs to obligate intracellular parasites and is able to invade all nucleated cells in warm-blooded animals including humans [142]. *T. gondii* infection upregulates and maintains the ZBP1 expression [143,144]. Deletion of ZBP1 promotes the replication rate of *T. gondii* in macrophages or mice and reduces host survival against oral challenge, confirming the suppression of ZBP1 on *T. gondii* infection [145]. But the deletion of ZBP1 does not influences the cell viability, and increases the level of proinflammatory cytokines, such as TNFα, IL-6, and MCP-1 in infected mice or BMDMs, showing ZBP1 is not involved in the inflammatory cell death during *T. gondii* infection [145,146]. Although BMDMs lacking ZBP1 present lower nitric oxide concentration (NO, an important immune molecule), a recent study confirmed that nitric oxide synthase (iNOS) lacking mice completely resist to *T. gondii* infection [145,147]. Thus, the relevance of ZBP1 to *T. gondii* infection could need more effort to illuminate.

## 6. ZBP1 Signaling in Auto-Inflammatory Diseases

Sensing host-derived nucleic acid ligands causes the spontaneous activation of ZBP1, that is responsible for many auto-inflammatory diseases, such as perinatal lethality, inflammatory bowel disease (IBD), heatstroke and oxidative stress-induced injuries, and acute pancreatitis (AP). 

In the setting of pathogens infection, RIPK1 is an important member of the ZBP1-PANoptosome and the key to ZBP1-mediated PANoptosis [23,57]. Paradoxically, various genetic and biochemical approaches have confirmed that the RHIM not kinase activity of RIPK1 disturbs the ZBP1-RIPK3 necrosome and avoids the aberrant necroptosis in the absence of infection (Figure 4). RIPK1^−/−^ mice or RIPK1^mR/mR^ mice expressing RIPK1 with mutated RHIM develop systemic inflammation and early postnatal lethality due to ZBP1-mediated necroptosis in multiple tissues [148,149,150,151]. RIPK1^E-KO^ mice (with the epidermis-specific RIPK1-knockout) suffer from skin inflammation because of ZBP1-dependent keratinocyte necroptosis [149,152]. The dendritic cell (DC)-specific RIPK1^−/−^ mice with loss of RIPK1 kinase activity in other cells, exhibit spontaneous colonic inflammation relied on ZBP1-RIPK3 necrosome [153]. Advanced research indicated that the Zα2 and RHIM1 of ZBP1 are vital for this RIPK1^−/−^ caused spontaneous necroptosis, and endogenous retroviruses (ERV)-derived complementary reads were detected in epidermal RNA [15,58,154] (Figure 4). Besides that, a histone methyltransferase safeguarding genome stability, SETDB1 deficiency in intestinal stem cells (IECs) liberates the ERV and products excessive viral mimicry [155]. The Zα and RHIM1 domains-dependent ZBP1 sense the viral dsRNA to induce necroptosis in IECs, which inevitably destroys the homeostasis of the epithelial barrier and causes inflammatory bowel disease (IBD) [155]. Hence, excluding pathogenic nucleic acid ligands, endogenous ERV-derived dsRNA is also effectively recognized by ZBP1 that induces spontaneous cell death, autoimmune inflammatory diseases, and even death.

In addition to RIPK1, ADAR1 is another host antagonist protein against ZBP1 signaling. ADAR1 is responsible for the adenosine to inosine (A-to-I) edits of ERV-derived dsRNA and represses abnormal immune responses activated by ERVs [156,157,158,159]. Deletion of ADAR1 or mutant of the Zα domain of ADAR1 causes the accumulation of ERV-derived dsRNA and spontaneous activation of ZBP1 [160,161]. The abnormal activation of ZBP1 causes embryonic lethality, intestinal cell death and skin inflammation similar to RIPK1^−/−^ or RIPK1^mR/mR^ mice [162,163]. ZBP1 plays a dual role in mice with impaired ADAR1 function. On the one hand, ZBP1 induces RIPK3-mediated necroptosis and CASP-8-mediated apoptosis which is MAVS-independent [162,163]. On the other hand, ZBP1 promotes the RIPK1- and RIPK3-independent, but MAVS-dependent IFN-I responses and stimulates interferonopathies [160].

Impaired mitochondrial DNA (mtDNA) is another endogenous ligand to activate ZBP1 (Figure 4). PUMA, NF-κB signaling induced proapoptotic protein, facilitates the rupture of mitochondrial membrane, the cytosolic release of mtDNA, and mtDNA-ZBP1 binding, leading to necroptosis in CASP-inactivated HT-29 cells [164,165,166]. A lack of FADD/CASP8 in IECs leads to ZBP1-dependent IBD, but deletion of PUMA can rescue embryonic death via reducing mtDNA-ZBP1 binding in FADD^−/−^ mice [164,167]. Beyond that, mtDNA-ZBP1 interaction mediated inflammation also is the result of chronic oxidative stress. The chronic low-level oxidative stress-stimulated retinal pigment epithelial (RPE) cells show mtDNA fragmentation and cytosolic release, and the damaged mtDNA is packed into extracellular vesicles (EVs) that transfer to surrounding microglia. The cytoplasmic mtDNA-ZBP1 binding in RPE cells and microglia triggers the TBK1/IRF3-dependent the expression of proinflammatory cytokines, such as IL-1α/β, IL-8, and TNFα, which contributes to the aberrant activation of microglia and ocular pathologies [168]. The ZBP1-mtDNA complex-mediated inflammation in response to oxidative stress also is demonstrated in lung epithelial cells [169].

Intriguingly, the latest research indicated that the ZBP1 without binding to any nucleic acid ligands can also provoke cell death when suffering from heatstroke [9]. Heat stress upregulates ZBP1 via HSF1 and promotes PANoptosome assembly resulting in PANoptosis in BMDMs and circulatory failure, organ injury, and lethality in mice. The Zα2 of ZBP1 is unnecessary but RHIM1 of ZBP1 is pivotal to interact with RIPK3. Lacking ZBP1 or RIPK3 significantly relieved alleviates the fatal injuries from extreme hyperthermia [9].

Moreover, the uncontrolled expression of ZBP1 and following ZBP1-dependent pyroptosis are the main reason for acute pancreatitis (AP) [170]. The tRF3-THr-AGT, an endogenous tRNA-derived small RNAs (tsRNAs), specifically binds the 3’ untranslated regions (3’-UTR) of ZBP1 mRNA to form short dsRNA that causes the degradation of ZBP1 mRNA [170]. In the cellular and animal AP models, the tRF3-THr-AGT is aberrantly decreased resulting in the upregulation of ZBP1, the activation of ZBP1-mediated NLRP3 inflammasome, and the extracellular release of cytokines and DAMPs [170,171]. In addition, further verification is needed to affirm whether the upregulated ZBP1 mediates apoptosis or necroptosis in acinar cells to aggravate AP.

## 7. ZBP1 Signaling in Tumor Immunity

Although ZBP1-induced aberrant cell death and inflammation are harmful to the host, there is a positive effect in the background of tumorigenesis. Advanced solid tumors generally suffer from necroptosis [172]. However, the tumor cell necroptosis may not be related to the TNFα-RIPK1 pathway [173,174]. In the late breast cancer cells, glucose deprivation causes the cytosolic release of mtDNA through mitochondrial protein NOXA. The ZBP1-mtDNA binding initiates RIPK3-MLKL-dependent tumor necroptosis and blocks metastasis in the lung [174]. In irradiated MC38 tumors, the ZBP1-mediated necroptotic signaling is crucial for anti-tumor immunity. That is not only because of ZBP1-mediated tumor necroptosis but also attribute to mtDNA release via the necroptotic signaling. The cytoplasmic mtDNA activates the cGAS-STING pathway and IFN-I responses. Deletion of ZBP1 inhibited the activation of cGAS-STING signaling and priming of CD8^+^ T cells in tumor cells with radiation treatment [175]. Mechanistically, the cytoplasmic release of mtDNA induced by ZBP1-necroptosis could be attributed to the translocation of RIPK3-MLKL necrosome to mitochondria. On the one hand, RIPK3 targets and activates the mitochondrial pyruvate dehydrogenase complexes, which require RIPK3-MLKL to be recruited into the mitochondria. Mechanistically, the cytoplasmic release of mtDNA induced by ZBP1-necroptosis could be attributed to the translocation of RIPK3-MLKL necrosome to mitochondria. On the one hand, RIPK3 targets and activates the mitochondrial pyruvate dehydrogenase complexes, which require RIPK3-MLKL to be recruited into the mitochondria [176]. On the other hand, the mitochondria membrane contains phospholipids, including phosphatidylinositol phosphates that anchor MLKL to the plasma membrane and nuclear membrane [23,177,178,179,180]. For these reasons, the hypothesis that RIPK3-MLKL necrosome punch pores in the mitochondrial membrane leading to mtDNA divulgence is possible. Nevertheless, it is worth affirming that the release of mtDNA caused by the ZBP1-mediated necroptotic signaling cascade and subsequent IFN-I responses via the cGAS-STING pathway will shape strong positive feedback of antitumor immunity [175,181].

In addition to mtDNA, the endogenous dsRNA remains the important ligand for ZBP1-mediated tumor immunity. However, the ZBP1 sensing for endogenous dsRNA is still interdicted by ADAR1. The latest research points out that some 3’UTR of ISGs mRNA containing short interspersed nuclear elements (SINEs) or GU-type simple repeats and naturally fold into Z-type dumbbells [161]. Such Z-RNA is segregated by the Zα domain of cytoplasmic ADAR1-p150 isoform or accumulates to activate ZBP1-dependent tumor necroptosis when ADAR1 is depleted or mutated [161] (Figure 4). The deficiency of ADAR1 encouraged ZBP1-mediated PANoptosis and limited tumorigenesis [33] (Figure 4). The antagonism strategy of ADAR1 against ZBP1 is very similar to the E3 of VACV, and all block the recognition of ZBP1 for activated Z-RNA ligands through the twin Zα domains [6,33]. ADAR1’s restriction on ZBP1 signaling complements the mechanism of the oncogenic role of ADAR1, and developing new drugs capable of repressing ADAR1 function or directly activating ZBP1 may be the new strategy for treating cancer [33,161,182]. 

Through genetic engineering, previous studies have applied ZBP1 to tumor therapy. Various genetic adjuvants enhance vaccine immunogenicity and strengthen adaptive immune responses [183,184,185,186]. Co-inoculation of the plasmid expressing ZBP1 (pZBP1) and the DNA vaccine encoding immunogenic tumor-associated antigens (TAAs) in mice more effectively activated NF-κB and IFN-I signaling and enhanced the TAA-specific cytotoxic CD8^+^ T lymphocytes (CTLs) and CD4^+^ Th1 responses, compared to the empty vector plus TAAs vaccine inoculation [187]. Compared to the solely vaccinated mice with pTRP2 (tyrosinase-related protein-2, a highly expressed glycoprotein in human melanomas), the higher percentage of TRP2-specific IFN-γ-producing CD8^+^ T cells and more durable stronger resistance to B16 melanoma was detected in co-vaccinated mice with pZBP1 and pTRP2 [187]. In addition, modifying oncolytic viruses to express therapeutic genes is a significant tumor immunotherapy strategy [188]. A ZBP1-armed oncolytic vaccinia virus has been created as a fast and effective self-recognition immune enhancement system. The VACV expressing ZBP1 induces T cell infiltration (especially TAA-specific T cells) which establishes a powerful antitumor immune response system and strongly controls tumor volume [189]. Although the above two experiments did not emphasize the relationship between ZBP1 and tumor cell death, the ZBP1-mediated antitumor therapeutic effect has been confirmed in vivo.

## 8. Concluding Remarks

ZBP1 is another Z-NA binding protein in addition to ADAR1 in mammals. Initially, ZBP1 was shown to be a cytoplasmic DNA sensor that promotes innate immune responses, and subsequent studies confirmed the importance of ZBP1 in PANoptosis initiation. ZBP1 signaling typically relies on the combination of the Zα2 domain with active ligands, such as virus-derived Z-RNA (VACV and IAV) [6,23,161,190]. In addition to sensing invasive pathogens, ZBP1 also binds with endogenous nucleic acid ligands, such as ERV-derived dsRNA and cytoplasmic mtDNA [155,168]. In surprise, endogenous ERV-derived Z-RNA recently has been confirmed to be present in some 3’UTR of ISGs mRNA that contain inverted SINEs or GU-type simple repeats and is sensed by ZBP1 in the absence of ADAR1-p150 [161]. In addition, the histone chaperone inhibitor, CBL0137 can directly induce Z-DNA formation in the LINE-1 element and activate ZBP1 in the presence of ADAR1-p150 [161]. The ZBP1 agonists will be a potential therapeutic avenue for tumors with limited ZBP1 function. However, there is no direct evidence as to whether the cytoplasmic damaged mtDNA detected by ZBP1 belongs to the Z-conformation. Moreover, even without ligands binding function, ZBP1 also acts as an adapter protein of AIM2 or TLR3/4-TRIF to transfer PANoptosis signaling [17,84].

ZBP1-PANoptosome initiates PANoptosis and plays a double-edged sword role in anti-infection and development. ZBP1-mediated PANoptosis efficiently limits the replication of intracellular pathogens, and stimulates the recruitment of immune cells and adaptive immune responses, which is beneficial to host survival [4]. Nonetheless, the excessive cell death activated by ZBP1 also provokes lethal inflammatory cytokine storm, organ dysfunction even host mortality. For instance, ZBP1-dependent PANoptosis restrict the viral loads of IAV and SARS-CoV-2 but caused severe inflammation and lung dysfunction in critical patients [23,95]. ZBP1-mediated brain cell death may be a key cause of the neuropathology during HSV-1 or SARS-CoV-2 infection [104,125]. In addition, the endogenous activation of ZBP1 causes multiple auto-inflammatory diseases. ZBP1 binding with ERV-derived dsRNA activates embryonic lethality in RIPK1^−/−^ mice and colitis in RIPK1^−/−^ IECs [58,155], and binding with leaked mtDNA leads to chronic oxidative stress-induced ocular pathologies [168]. However, the endogenous activation of ZBP1 also has a positive effect in the setting of cancer. ZBP1-PANoptosome promotes tumor death and immune cell infiltration, and limits tumor volume and metastasis [33,174]. 

Both the pathogen and the host have evolved skills to regulate ZBP1 signaling to ensure pathogen replication or immune homeostasis (Table 1 and Table 2). Relying on the homogenous Zα domains, ADAR1-p150 and VACV E3 segregate the activated Z-RNA of ZBP1 to inhibit ZBP1 signaling [6,33,161]. RIPK1, MCMV M45, HSV-1 ICP6, and VZV ORF20 all block the signaling transduction of ZBP1-RIPK3 via shared RHIM domains [4,85,89,150]. Furthermore, SARS-CoV-2 ORF8 is a new negative regulator that antagonizes the IFN-γ-stimulated expression of ZBP1 [96]. ZBP1 may also play a role for others viruses, and its influence could be antagonized by some unknown virulence factors which need follow-up studies to investigate. 

In this review, we have elucidated the ZBP1-mediated signaling mechanisms of innate immune response and PANoptosis. We also discussed the pathogens and host-developed multiple regulation strategies for ZBP1-mediated signaling; ZBP1 plays a double-effective role in infection and host health. Understanding the impact of ZBP1 signaling on host or pathogens and developing novel positive or negative regulated drugs against ZBP1 signaling will be helpful to anti-infection, alleviation of inflammatory disease, and anti-tumor immunity.

## Figures and Tables

**Figure 1 ijms-23-10224-f001:**
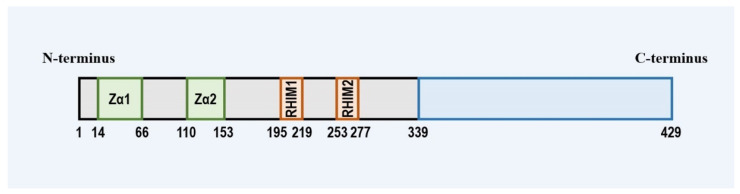
Illustration of the domain structure of ZBP1. ZBP1 consists of two N-terminal Z-form nucleic acid binding domains, two RHIM domains of the intermediate segment, and the C-terminal domain.

**Figure 2 ijms-23-10224-f002:**
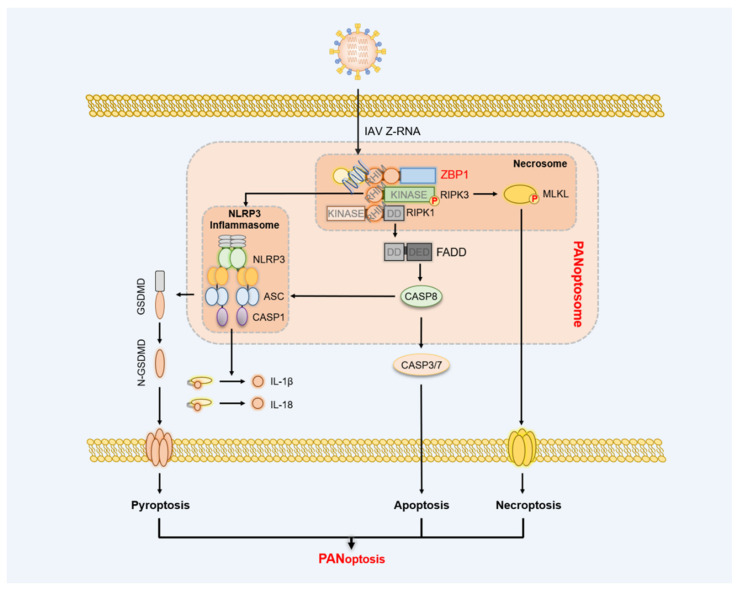
The pattern of ZBP1-initiated PANoptosome assembly. The sensing of ZBP1 Zα for activated ligands allows ZBP1 interaction with RIPK1 and RIPK3 via RHIM domains and forms the core scaffolds of PANoptosome. RIPK3 induces the phosphorylation and membrane translocation of MLKL, which is the hallmark of necroptotic signaling. In addition, the homotypic interaction between RIPK1, FADD, and CASP8 liberates the cleaving activity of CASP8 and CASP3/7 that causes apoptosis. ZBP1-mediated apoptosis and necroptosis signaling promote the assembly and activation of NLRP3 inflammasome. Activated NLRP3 inflammasome matures GSDMD, IL-1β, and IL-18, named pyroptosis.

**Figure 3 ijms-23-10224-f003:**
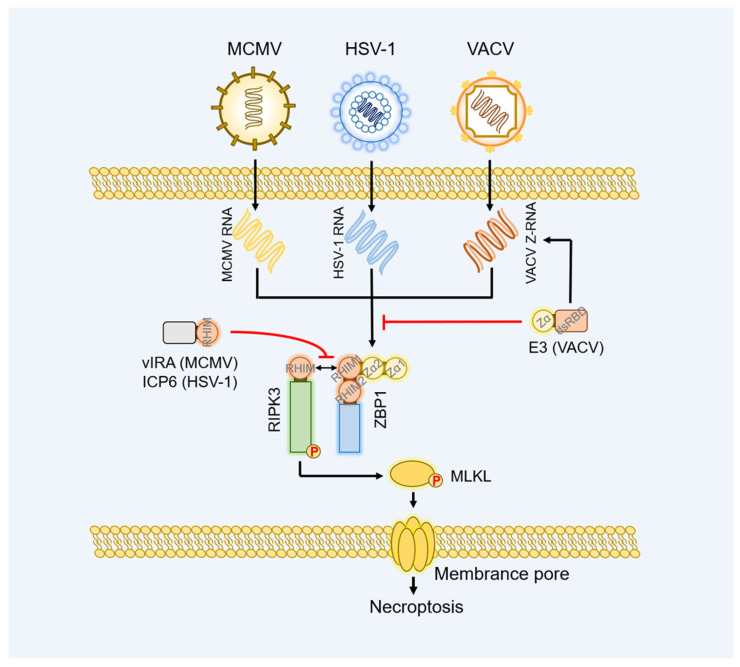
DNA viruses developed antagonistic mechanisms against ZBP1 signaling. After sensing virus-derived RNA, ZBP1 triggers necroptosis via the ZBP1-RIPK3-MLKL axis against MCMV, HSV-1, or VACV infection. However, the ZBP1 signaling is blocked by the DNA viruses-encoded virulence factors. At the beginning of the necroptosis signaling, the VACV E3 protein shields the VACV Z-RNA by its Zα domain and abolishes the activation of ZBP1. After that, the homologous RHIM of MCMV vIRA and HSV-1 ICP6 destroys the signaling transduction from ZBP1 to RIPK3.

**Figure 4 ijms-23-10224-f004:**
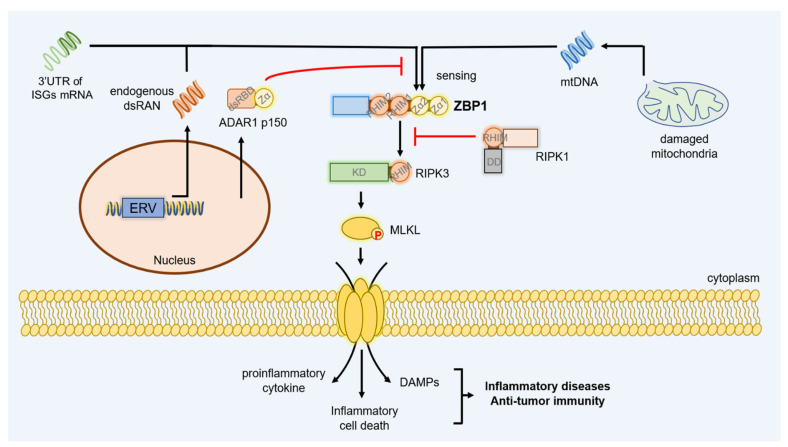
The model for endogenous ZBP1 activation and the host antagonistic mechanisms targeting ZBP1 signaling. The endogenous retroviruses (ERV)-derived dsRNA, released cytoplasmic mitochondria DNA (mtDNA), and Z-RNA enriched in 3’UTR of ISGs mRNA that harbor inverted SINEs or GU-type simple repeats act as the activated ligands of ZBP1 and induce endogenous ZBP1 signaling. The endogenous activation of ZBP1 causes many auto-inflammatory diseases or stimulates tumor immunity. ADAR1-p150 inhibits the ligands sensing of ZBP1 by the Zα domain and RIPK1 blocks the signaling transduction of ZBP1 by its RHIM.

**Table 1 ijms-23-10224-t001:** Programed cell death mediated by ZBP1 during microbial infection.

Pathogens		Genome or Type	Ligands of ZBP1	Type of Cell Death	Subcellular Location of ZBP1	Virulent Factor	Antagonism Mechanism against ZBP1 Signaling	Reference(s)
Viruses	MCMV	dsDNA	nascent RNA	Necroptosis	cytoplasm	vIRA	Disrupts ZBP1–RIPK3 interactions	[79,80,81,82,83]
	HSV-1	dsDNA	nascent RNA	PANoptosis	cytoplasm	ICP6	Disrupts ZBP1–RIPK3 interactions	[84,85,86,87,88]
	VZV	dsDNA	nascent RNA	Apoptosis	cytoplasm	ORF20	Disrupts ZBP1–RIPK3 interactions	[89]
	VACV	dsDNA	VACV Z-RNA	Necroptosis	cytoplasm	E3	Interdicts the sensing of ZBP1 for viral Z-RNA	[6,90,91]
	IAV	ssRNA (−)	IAV Z-RNA	PANoptosis	nucleus			[23,61,92]
	WNV	ssRNA (+)						[93]
	ZIKV	ssRNA (+)		Necroptosis				[94]
	SARS-CoV-2	ssRNA (+)		PANoptosis		ORF8	Inhibits the IFN-γ-induced expression of ZBP1	[95,96]
Bacterium	*Escherichia coli;* *Yersinia pseudotuberculosis;* *F. novicida.*	Gram-negative		PANoptosis		EspL	Cleaves RHIM fragments of all RHIM-containing host proteins	[84,97]
Fungus	*Candida albican; Aspergillus fumigatus*			PANoptosis				[98]

**Table 2 ijms-23-10224-t002:** The host proteins regulating the ZBP1 signaling.

Host Regulatory Factors	Positive(P) or Negative(N) Regulation	Regulatory Mechanisms	Reference(s)
AIM2	P	Interacts with ZBP1 to drive PANoptosis	[84]
TRIM34	P	Induces the K63-linked polyubiquitination of ZBP1 on the K17 position	[116]
CASP6	P	Strengthens ZBP1–RIPK3 interactions	[119]
PUMA	P	Promotes the cytoplasmic release of mtDNA and activation of ZBP1	[164]
ADAR1	N	Inhibits the ZBP1 sensing for activated Z-RNA	[33,160,161,162,163]
RIPK1	N	Disrupts ZBP1–RIPK3 interactions	[149,150]
SETDB1	N	Holds down the level of ERV-derived dsRNA and activation of ZBP1	[155]
tRF3-Thr-AGT	N	Degrades the mRNA of ZBP1	[170]

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
