# Peer review of "ZBP1: A Powerful Innate Immune Sensor and Double-Edged Sword in Host Immunity"

_ijms, 2022, doi:10.3390/ijms231810224_

Round 1

Reviewer 1 Report

The review is comprehensive and generally well-written. However there are some issues that I think should be addressed.

  • CASP6 results cited have not been reproduced by others
  • Panoptosis is a poorly defined concept as different tissues die in one way or the other but do not use multiple choice to decide how. The author use the word panoptosis when they should correctly use necroptosis or apoptosis.
  • The author might want to include the role of Z-DNA in maintaining latent EBV infection (PMID 35328502)
  • For a possible role of NSP13 and Z-RNA in coronaviruses see also (PMID: 35784331)

Minor corrections: 

Line 16 - succedent

121 - absentation

525 - antagonism mechanism

555 unknow

In table2 - to further THE assemble of PANoptosis

Reviewer 2 Report

The review paper “ZBP1: A Powerful Innate Immune Sensor and Double-Edged Sword In Host Immunity” by Hao et al. described the mechanisms of ZBP1 signaling, and various antagonistic strategies of host and pathogen against ZBP1 based on these mechanisms.

The topic is important, and the manuscript is well written containing the latest research results and papers. Therefore, I recommend publication after minor revision.

Please cite the following crucial references in the appropriate sections of your manuscript with a one-sentence description.

Proc Natl Acad Sci U S A. 2022 Jun 14;119(24):e2113872119.
ZBP1 promotes inflammatory responses downstream of TLR3/TLR4 via timely delivery of RIPK1 to TRIF
PMID: 35666872

Nature. 2022 Jul;607(7920):784-789.
ADAR1 prevents autoinflammation by suppressing spontaneous ZBP1 activation
PMID: 35859175

Nature. 2022 Jul;607(7920):776-783.
ADAR1 averts fatal type I interferon induction by ZBP1
PMID: 35859176

Nature. 2022 Jul;607(7920):769-775.
ADAR1 mutation causes ZBP1-dependent immunopathology
PMID: 35859177

Curr Opin Virol. 2021 Dec;51:134-140.
Viral Z-RNA triggers ZBP1-dependent cell death
PMID: 34688984

Trends Immunol. 2018 Feb;39(2):123-134.
ZBP1: Innate Sensor Regulating Cell Death and Inflammation
PMID: 29236673

Minor comment
Page numbers of some references in the manuscript are missing.
e.g.
Line 656, Cold Spring Harb Perspect Biol 2019, 11, (5). → Cold Spring Harb Perspect Biol 2019, 11, (5), a032813.
Line 661,  J Cell Sci 2021, 134, (10). → J Cell Sci 2021, 134, (10), jcs258446.
Line 762, Sci Immunol 2016, 1, (2). → Sci Immunol 2016, 1, (2), aag2045.
Line 797, Sci Adv 2020, 6, (47). → Sci Adv 2020, 6, (47), eabc3465.
Line 826, EMBO Rep 2019, 20, (2). → EMBO Rep 2019, 20, (2), e46518.
Line 879, Int J Mol Sci 2018, 19, (7). → Int J Mol Sci 2018, 19, (7), 2065.
Line 898, Pathogens 2022, 11, (2). → Pathogens 2022, 11, (2), 257.
Line 923, Viruses 2019, 11, (2). → Viruses 2019, 11, (2), 162.
Line 959, Infect Immun 2019, 87, (5). → Infect Immun 2019, 87, (5), e00024-19.
Line 963, Infect Immun 2021, 89, (5). → Infect Immun 2021, 89, (5), e00021-21.
Line 987,  J Exp Med 2020, 217, (7). →  J Exp Med 2020, 217, (7), e20191913.
